# Metabolomic Insights into Wild and Farmed Gilthead Seabream (*Sparus aurata*): Lipid Composition, Freshness Indicators, and Environmental Adaptations

**DOI:** 10.3390/molecules30040770

**Published:** 2025-02-07

**Authors:** Frutos C. Marhuenda-Egea, Pablo Sanchez-Jerez

**Affiliations:** 1Department of Biochemistry and Molecular Biology and Agricultural Chemistry and Edafology, University of Alicante, Carretera San Vicente del Raspeig s/n, 03690 Alicante, Spain; 2Department of Marine Sciences and Applied Biology, University of Alicante, Carretera San Vicente del Raspeig s/n, 03690 Alicante, Spain; pablo.sanchez@ua.es

**Keywords:** HR-MAS NMR, lipid metabolism, aquaculture, wild fish, metabolic trade-offs, K-value

## Abstract

Background/Objectives: This study explores the metabolic adaptations and quality differences between wild and farmed gilthead seabream (*Sparus aurata*), with a particular focus on lipid composition and metabolite profiles. These differences are examined in the context of varying environmental conditions, feeding regimes, and post-harvest processes. High-resolution magic-angle-spinning nuclear magnetic resonance (HR-MAS NMR) spectroscopy was employed to perform the metabolomic analysis. Results: Farmed seabream exhibited higher lipid content and PUFA levels (e.g., DHA and EPA) due to aquaculture diets, while wild seabream showed lower lipid concentrations and elevated levels of polar metabolites. Metabolic trade-offs in wild seabream reflected greater physical activity and environmental adaptation. The K-value indicated faster spoilage in farmed seabream, particularly from Greece, linked to handling conditions. HR-MAS provided precise, reproducible results, allowing direct quantification of key metabolites without altering sample integrity. Methods: HR-MAS NMR was employed to analyze muscle tissue from wild and farmed seabream (produced in Spain and imported from Greece), providing high-resolution spectra without requiring sample extraction. Metabolite quantification included polyunsaturated fatty acids (PUFAs), creatine, taurine, lactate, and trimethylamine N-oxide (TMAO). Freshness was monitored using the K-value index, calculated from ATP derivative levels in samples stored at 4 °C. Conclusions: The study highlights the influence of diet and environment on the metabolic profiles of seabream. HR-MAS NMR emerges as a robust method for metabolomic studies and freshness assessment. Findings emphasize the potential for dietary adjustments to optimize aquaculture practices and fish quality while underscoring the importance of sustainable production strategies. Further research into lipid metabolism genes and environmental factors is recommended to deepen understanding of these adaptations.

## 1. Introduction

The Mediterranean Sea has long been a cornerstone for marine biodiversity and a hub for fisheries and aquaculture activities [1]. Among its valuable marine species, the gilthead seabream (*Sparus aurata*) holds a significant position due to its ecological and economic importance. In 2022, total aquaculture production of seabream in Europe and the Mediterranean was estimated at 320,630 tonnes, representing a 1.8% increase compared to 2021. The estimated first-sale value reached 1574.8 million euros [2].

Traditionally a target of artisanal fisheries, the increasing demand for seabream in global markets has propelled the growth of marine aquaculture, making it a focal species in Mediterranean aquaculture. The cultivation of seabream has evolved into a highly sophisticated industry, leveraging advances in breeding, nutrition, and environmental monitoring to optimize production and sustainability [3].

The natural distribution of seabream, which thrives in coastal and estuarine habitats across the Mediterranean, has historically supported robust fisheries. These fisheries, while culturally ingrained and economically vital, face growing challenges such as overfishing, habitat degradation, and climate change. As a response, aquaculture has emerged as a sustainable alternative to supplement the declining yields from wild stocks [4]. Currently, gilthead seabream ranks among the most farmed fish species in the Mediterranean, with production concentrated in countries such as Greece, Turkey, and Spain [1]. The ability of this species to tolerate a wide range of salinities and temperatures has further solidified its role in the expansion of marine aquaculture.

The evolution of seabream aquaculture, from traditional net pens to modern recirculating aquaculture systems (RASs), reflects the industry’s commitment to improving yield and environmental stewardship. Early farming practices, characterized by extensive systems with limited control over environmental variables, often faced challenges such as disease outbreaks and inefficient feed conversion. However, advances in genetic selection, feed formulation, and water quality management have revolutionized the sector. Modern systems now incorporate real-time monitoring technologies and biosecurity protocols, reducing the environmental footprint and enhancing fish welfare [1].

### 1.1. Benefits and Drawbacks of Marine Aquaculture in the Mediterranean

Marine aquaculture offers numerous advantages, including the ability to meet growing seafood demand without exacerbating pressures on wild fish populations. The cultivation of seabream provides high-quality protein with a favorable fatty acid profile, contributing to food security and human health. Additionally, the economic benefits extend to job creation and rural development in coastal communities [1].

However, the industry is not without its drawbacks. Concerns over nutrient enrichment, the spread of diseases, and the potential for genetic introgression with wild populations persist. Regulatory frameworks and adherence to sustainability certifications have sought to mitigate these impacts, but challenges remain, particularly in regions with less stringent enforcement. Moreover, the reliance on fishmeal and fish oil in seabream diets raises questions about the sustainability of feed sources, prompting research into alternative ingredients such as insect meal and algae-based oils [3].

### 1.2. Methodologies and Decision-Making Factors in Seabream Aquaculture

Research and development in seabream aquaculture has focused on optimizing production efficiency while minimizing environmental impacts. Experimental studies have evaluated stocking densities, feed formulations, and water quality parameters to establish best practices. The use of integrated multi-trophic aquaculture (IMTA) systems, where seabream is co-cultured with species such as seaweed and shellfish, has shown promise in enhancing nutrient recycling and reducing environmental impacts [3,5].

Decision-making in the industry often involves balancing economic viability with ecological considerations. Factors such as site selection, water quality, and regulatory compliance play critical roles in the success of aquaculture operations. Additionally, consumer preferences and market trends influence product differentiation strategies, such as organic certification or valorization of niche markets (https://www.iemed.org/publication/aquaculture-in-the-mediterranean, accessed on 4 February 2025).

### 1.3. Advantages and Limitations of HR-MAS in Fish Studies

In recent years, metabolomics and lipidomics have become essential tools for understanding biochemical processes in aquatic species, providing valuable insights into their physiological states and responses to environmental stressors [6]. High-resolution magic-angle-spinning (HR-MAS) nuclear magnetic resonance (NMR) spectroscopy has emerged as a particularly valuable technique in fish studies due to its non-destructive nature, enabling the direct analysis of complex biological matrices while preserving tissue integrity. This approach facilitates the evaluation of spoilage-related biochemical changes during storage, addressing critical food safety and quality concerns [7].

HR-MAS bridges gaps in metabolomics by allowing in situ analysis of intact tissues, offering both qualitative and quantitative data with exceptional sensitivity. Unlike traditional techniques requiring extensive sample preparation, HR-MAS minimizes artifacts and retains the native biochemical state. Early applications of HR-MAS focused on identifying metabolites and lipids, but advancements in pulse sequences and spectral resolution have expanded its utility to include kinetic spoilage analyses and the evaluation of storage conditions [7].

The method’s ability to simultaneously analyze hydrophilic and hydrophobic metabolites makes it versatile for fisheries science. However, limitations such as high instrumentation costs, specialized expertise requirements, and reduced sensitivity to low-abundance compounds compared to mass spectrometry may hinder routine adoption. Despite these challenges, HR-MAS has proven effective in monitoring oxidative stress markers, lipid oxidation, and other quality deterioration indicators, enabling comparative analyses of preservation methods like freezing and vacuum packaging. These insights directly benefit the seafood industry by optimizing storage practices and ensuring product safety [8,9].

The cultivation of gilthead seabream in the Mediterranean highlights aquaculture’s potential to address food security challenges while supporting regional economies. Sustainable practices, including advanced technologies and ecosystem-based approaches, are essential for mitigating environmental impacts and ensuring the sector’s long-term viability. HR-MAS NMR spectroscopy offers a robust framework for studying seabream metabolism and spoilage dynamics, advancing quality control and food safety efforts. Future research should focus on integrating HR-MAS with complementary techniques to achieve a comprehensive understanding of fish spoilage, promoting innovation in fisheries science [10].

The objective of this study is to investigate the metabolic and biochemical adaptations of the gilthead sea bream (*Sparus aurata*) under varying environmental and nutritional conditions, utilizing high-resolution magic-angle-spinning (HRMAS) NMR spectroscopy. By characterizing the metabolomic profiles of muscle tissue, this research aims to identify biomarkers associated with growth performance, stress response, and overall health status. Particular emphasis is placed on evaluating the condition factor (K) as a critical parameter linking metabolic alterations to the fish’s somatic growth and welfare [7]. Additionally, the study seeks to assess the impact of specific dietary formulations and environmental stressors on the fish’s physiological state, with the ultimate goal of optimizing aquaculture practices for sustainable and efficient production of *S. aurata*.

## 2. Results

### 2.1. Morphological Characteristics of the Fish

Wild gilthead seabream had an average weight of 288.3 ± 35.9 g, whereas those farmed in aquaculture along the Spanish coasts had an average weight of 510.4 ± 95.6 g, and those farmed along the Greek coasts had an average weight of 501.0 ± 23.4 g. The Total Length (TL), measured from the tip of the snout to the end of the longest caudal (tail) fin lobe, with the lobes compressed along the midline, was 28.0 ± 1.3 cm, 30.1 ± 1.6 cm, and 29.7 ± 0.3 cm for wild seabream, those farmed along the Spanish coasts, and those farmed along the Greek coasts, respectively. The Fork Length (FL), measured from the tip of the snout to the fork of the caudal fin, was 26.0 ± 0.9 cm, 28.5 ± 0.9 cm, and 28.0 ± 0.5 cm for wild seabream, those farmed along the Spanish coasts, and those farmed along the Greek coasts, respectively. The weight decreases in wild seabream compared to farmed ones was 42.5% (farmed in Greece) to 43.5% (farmed in Spain), while the length difference ranged from 7% to 8%.

### 2.2. Analysis of HR-MAS Spectra

Figure 1 presents the 1H-HRMAS spectrum at 500 MHz of muscle samples from wild gilthead seabream and farmed gilthead seabream raised in marine aquaculture systems. The assignments were validated against published spectra of standards available in databases [8,9] (HMDB; https://hmdb.ca/, accessed on 4 February 2025), as well as other published studies involving fish muscle or fish oil samples, using different chemometrics tools [11,12,13,14,15,16,17,18,19,20,21,22].

The samples from three individuals of each type (wild, farmed in Spain, and farmed in Greece) demonstrated excellent reproducibility, both during the initial analysis and throughout cold storage at 4 °C. It is important to note that 1H-MAS is a direct measurement technique applied to muscle tissue, precluding the possibility of sample mixing [8].

The spectra obtained from the 2D homonuclear correlation experiments (COSY-HRMAS) (Figure 2) and the 2D heteronuclear (HSQC-HRMAS) experiments displayed good resolution, enabling accurate assignment of various signals (Table 1). Additionally, these spectra revealed differences between polar molecules (more abundant in wild gilthead seabream) and lipids (more prevalent in farmed seabream) (Figure 2 and Figure 3). A predominance of signals attributed to fatty acids and triglycerides can be observed (Figure 2), likely reflecting lipids present within the muscle cells of farmed seabream.

We employed robust Principal Component Analysis (robPCA) [13] to identify the key molecular signals that distinguish gilthead seabream based on their origin (wild or farmed). The pseudo-spectrum signals, corresponding to the loadings of Principal Component 1 (PC1, 95.38%) (Figure 4), indicate that positive signals are more intense in the 1D 1H-HRMAS NMR spectra of farmed fish samples from both Spain and Greece. Conversely, negative signals in the pseudo-spectrum suggest greater intensity in the 1D 1H-HRMAS NMR spectra of wild seabream samples. The loadings presented in Figure 4 are instrumental in differentiating between wild and farmed seabream, as illustrated in the insert of Figure 4, which displays the robust PCA score plot. The ability to analyze whole muscle samples using HR-MAS, without the need for fractionation into polar and apolar components—as required in other experimental approaches—enhances the capacity to discern compositional differences between wild and farmed seabream (Figure 5).

Figure 5 illustrates the histogram of the mean values of the integral of the area under the signal of the selected molecules (polar metabolites and lipids) with standard errors, highlighting differences between wild and farmed gilthead seabream. Wild seabream exhibited higher levels of polar metabolites such as TMAO, taurine, creatine, and phosphocreatine (Figure 3). Figure 3 magnifies the region of the 1D 1H-HRMAS NMR spectra of muscle tissue from wild gilthead seabream and farmed gilthead seabream between 3 and 3.5 ppm. This region contains the taurine triplets (S-CH_2_ at 3.27 ppm, and N-CH_2_ at 3.42 ppm) and the TMAO singlet (N-CH_3_ at 3.28 ppm). These metabolites are associated with the greater proportion of muscle mass observed in wild seabream compared to their farmed counterparts. The value of the integral of the TMAO (N-CH3) signal at 3.28 ppm was corrected by subtracting the value of the integral of the left peak of the taurine triplet (S-CH2) at 3.27 ppm, for each sample. Conversely, farmed seabream, influenced by their diet, displayed significantly higher levels of lipids (UFA, DUFA, and DHA) in muscle tissue (Figure 4 and Figure 5). One notable lipid is linoleic acid, which is a common component of the feeds used in marine aquaculture.

Moreover, the farmed seabream from Spanish and Greek coastal waters showed remarkable similarities in their composition of polar metabolites, lipids, as well as size and weight. In aquaculture, achieving fish with standardized sizes and weights suitable for commercial markets is a priority. This is accomplished through diets with highly similar compositions tailored to the same fish species. Consequently, farmed seabream from both regions exhibited not only comparable physical appearances but also analogous metabolomic and lipidomic profiles.

### 2.3. Deterioration Patterns in Samples by Origin

The K-value, also known as the freshness index, was widely used to monitor fish quality. It reflects the degradation of ATP and its derivatives, indicating freshness loss through hypoxanthine accumulation. The formula for the K-value is as follows:K = ([HxR] + [Hx])/([ATP] + [ADP] + [AMP] + [IMP] + [HxR] + [Hx]) × 100
where HxR is hypoxanthine riboside (inosine), Hx is hypoxanthine, and ATP, ADP, AMP, and IMP represent adenosine nucleotides and derivatives [7,23]. A low K-value (<20%) indicates fresh fish, while values above 50% suggest advanced spoilage. To calculate the K-value, the corresponding signals from the 1H-HRMAS spectra were integrated. These signals include the doublet of IMP at 6.16 ppm (CH-1 Ribose), the doublet of HxR at 6.11 ppm (CH-1 Ribose), the singlet of Hx at 8.19 ppm (CH-8), and the singlet of ATP/ADP/AMP at 8.49 ppm (CH-2 Purine). Samples were stored at 4 °C under conditions similar to typical household refrigeration. Under these storage conditions, K-values increased by up to 33% after 21 days for wild gilthead seabream, 35% for seabream farmed in Spanish coastal waters after 17 days, and 41% for seabream farmed in Greek coastal waters after the same 17-day period [7].

Initial K-values were 12.5% for wild seabream, 12.6% for seabream farmed in Spain, and 21.7% for seabream farmed in Greece. The slope values for K-value increase were 0.95, 1.30, and 1.10 for wild gilthead seabream, farmed seabream from Spain, and farmed seabream from Greece, respectively (Figure 6). The R^2^ values of the linear regressions for K-value variations were 0.98, 0.99, and 0.92 for wild seabream, Spanish farmed seabream, and Greek farmed seabream, respectively. These values were comparable to those reported in other studies, despite differences in fish species or analytical techniques [7]. The main distinction lies in the slope values, which, in our case, were slightly lower than those reported by other authors [7].

## 3. Discussion

### 3.1. Morfological Differences Between Fish

The weight decreases in wild seabream compared to farmed ones likely result from the controlled feeding and farming conditions in aquaculture, which are optimized to achieve commercial sizes more efficiently. These differences in morphology, particularly weight, may influence the observed lipid and metabolite profiles, as farmed seabream exhibits higher lipid content due to their diet, whereas wild seabream shows a greater proportion of polar metabolites associated with muscle activity.

Farmed gilthead seabream from Spain and from Greece exhibit very similar appearances, making it difficult to distinguish them through visual inspection. When their molecular compositions are analyzed using HR-MAS, a similar profile is also observed (Figure 4). This similarity is likely attributable to their comparable diets, primarily based on aquaculture feeds.

### 3.2. Advantages of HR-MAS Analysis

High-resolution magic-angle-spinning nuclear magnetic resonance (HR-MAS NMR) is a powerful tool in metabolomic studies, particularly for complex tissues like fish muscle. Its ability to analyze samples in their natural state, without requiring destructive extractions, ensures that metabolite concentrations remain unaltered, providing an accurate representation of tissue conditions. This capability has been particularly valuable in comparative studies, such as analyzing wild versus farmed gilthead seabream (Figure 1), shedding light on how environmental and dietary conditions influence metabolism [24]. Furthermore, the technique produces high-resolution spectra, enabling the simultaneous identification and quantification of metabolites, including those at low concentrations. Moreover, data obtained from HR-MAS are highly suited for multivariate statistical anal-yses, such as robust Principal Component Analysis (robPCA) (Figure 4) and orthogonal partial least squares discriminant analysis (OPLS-DA), which have been extensively demonstrated in various research domains. These statistical tools enable the identification of specific metabolic patterns and biomarkers (Figure 4), reinforcing HR-MAS’s critical role in advancing metabolomic research [25].

### 3.3. Preliminary Discussion on Lipid Composition of Wild and Farmed Gilthead Seabream

The lipid composition of farmed gilthead seabream shows notable similarities to that of farmed salmon [9], despite these being different species. This highlights the direct impact of formulated diets in marine aquaculture systems. These findings align with previous studies demonstrating how aquaculture diets, predominantly based on fish oils and meals rich in polyunsaturated fatty acids (PUFAs), significantly influence the lipid profile of farmed fish [26,27].

In particular, omega-3 fatty acids such as DHA (22:6 n-3) and EPA (20:5 n-3) constitute a significant proportion of PUFAs in farmed seabream. This similarity arises from the dietary ingredients used in both cases, which were designed to promote growth and meet market standards for nutritional quality [28].

Previous studies have shown that lipid accumulation in the myocytes of farmed fish was closely related to diet composition and physiological mechanisms regulating energy storage. These studies indicate that the high availability of PUFAs in lipid-rich diets promotes triglyceride accumulation in myocytes, potentially impacting muscle metabolism and filet quality [29].

Lipid accumulation in myocytes was regulated by several key genes involved in lipid metabolism. Among these, peroxisome proliferator-activated receptors (PPARs) play a crucial role. PPARγ promotes adipocyte differentiation and triglyceride accumulation, while PPARα regulates fatty acid oxidation, balancing energy storage and utilization. Additionally, fatty acid-binding protein 3 (FABP3) facilitates intracellular fatty acid transport to storage or utilization sites [30].

Sterol regulatory element-binding protein (SREBP-1c) stimulates the expression of lipogenic genes, including those involved in triglyceride and fatty acid synthesis. Genes like FASN (fatty acid synthase) and DGAT (acyl-CoA: diacylglycerol acyltransferase) contribute to lipid storage by catalyzing de novo synthesis and triglyceride assembly, respectively. Conversely, lipid oxidation is driven by genes such as CPT1 (carnitine palmitoyltransferase 1), which regulates fatty acid transport to mitochondria, and ACOX, which initiates oxidation in peroxisomes [31].

In contrast, wild seabream exhibits a significantly different lipid profile (Figure 1), characterized by a lower overall lipid concentration and a higher prevalence of polar metabolites such as carbohydrates, amino acids, osmolytes, and organic acids. This can be attributed to the diverse, protein-rich diet available in their natural environment, which supports a metabolism oriented toward different energy and functional demands. Appreciable amounts of triglycerides were not detected in wild seabream myocytes, supporting the hypothesis of reduced lipid storage due to a diet lacking the high oil levels found in aquaculture feeds [32].

The comparison between farmed and wild seabream underscores the determining influence of diet and highlights the specific metabolic adaptations of each environment. High levels of DHA and EPA in farmed seabream and salmon confirm that the lipid profile in aquaculture fish is directly related to dietary composition. In contrast, the lower concentrations of these compounds in wild seabream reflect limited access to marine PUFA sources in their natural diet and excellent adaptation to the natural environment [33].

These results have important implications for understanding metabolic mechanisms in marine fish and for the aquaculture industry. The potential to adjust the lipid profile of farmed fish through dietary modifications could optimize both production efficiency and final product quality. It is worth noting that excessive triglyceride accumulation in myocytes could pose severe metabolic issues for farmed fish [34].

### 3.4. Metabolic Trade-Offs in Gilthead Seabream

Gilthead seabream faces a series of metabolic trade-offs arising from its environment, diet, and physiological activity. These trade-offs reflect adaptations to optimize resource utilization, balancing energy storage with expenditure on critical activities such as locomotion, growth, and reproduction. As a pelagic species, seabream is an active swimmer, necessitating constant access to energy. This limits excessive lipid accumulation, favoring instead the rapid mobilization of stored fatty acids for oxidation. Excessive lipid storage could increase buoyancy, disrupting hydrodynamic balance during swimming [35].

The activation of genes such as SREBP-1c is moderated in seabream due to its natural diet, which is rich in essential fatty acids (PUFAs). This limits the need for de novo lipogenesis and prioritizes the mobilization of stored lipids for energy demands [36].

In wild seabream, variability in food availability can limit lipid storage, whereas farmed seabream, provided with a constant lipid-rich diet, achieves higher reproductive success [34]. During reproductive stages, seabream redirects significant energy reserves toward gonadal development, reducing energy available for somatic growth.

Environmental factors such as temperature and food availability in pelagic habitats cause fluctuations that activate stress responses mediated by cortisol. This diverts energy from anabolic processes, such as growth and reserve accumulation, to catabolic mechanisms for stress management [37]. These metabolic trade-offs highlight the complex physiological adaptations of gilthead seabream to its environment and feeding conditions.

### 3.5. Metabolite Profiles of Wild and Farmed Gilthead Seabream

Wild gilthead seabream exhibits significantly higher levels of metabolites such as creatine, taurine, lactate, and trimethylamine N-oxide (TMAO) compared to farmed seabream (Figure 1, Figure 2 and Figure 3). These compounds are intrinsically linked to muscle activity and tissue quality. In wild seabream, the healthy muscle, rich in these metabolites, reflects higher physical activity, in contrast to farmed seabream, whose muscle contains elevated levels of lipids stored in myocytes [27,38,39,40,41,42]. Creatine is essential for the rapid storage and release of energy during sustained muscle contractions. Dietary supplementation of creatine in gilthead seabream has been shown to improve muscle quality, indicating its vital role in energy metabolism [43]. Lactate indicates greater anaerobic activity in muscle tissue, which is often associated with higher physical exertion levels in wild fish. Elevated lactate levels in wild seabream suggest a metabolism adapted to burst swimming and escape responses.

TMAO concentration is up to ten times higher in wild seabream (Figure 3, Figure 4 and Figure 5). This compound (TMAO) has multiple functions. It acts as a compatible osmolyte, stabilizing proteins in high-osmotic-pressure environments, such as the pelagic habitat. TMAO is produced in the liver from TMA, a compound generated in the intestine by microbial activity. The intestinal microbiota of wild seabream is likely more diverse due to the absence of antibiotic treatments, favoring higher TMAO production. In farmed seabream, the use of antibiotics alters this microbiota, reducing the levels of this key compound [42,44]. TMAO enhances osmotic tolerance and is essential for protein stability. The low concentration of TMAO in farmed seabream could reflect reduced adaptation to the natural environment and diminished muscle quality. Conversely, in humans, excessive TMAO accumulation is associated with cardiovascular diseases, highlighting its dual role in different biological contexts [45,46].

Taurine plays a key role as an antioxidant and in stabilizing cell membranes, contributing to muscle tissue integrity. It neutralizes reactive oxygen species (ROS) and stimulates the activity of antioxidant enzymes such as superoxide dismutase (SOD) and catalase (CAT). Dietary taurine supplementation has been found to enhance antioxidant enzyme activity and immune response in fish, highlighting its importance in maintaining muscle health [47]. In wild seabream, elevated taurine levels correlate with greater antioxidant capacity, enabling them to cope with oxidative stress associated with intense physical activity and environmental fluctuations. Diets in aquaculture systems often lack sufficient taurine, potentially leading to reduced antioxidant capacity in farmed seabream. This is also associated with increased susceptibility to oxidative damage and inferior muscle quality. Taurine also protects against oxidative damage caused by environmental pollutants and overcrowding in farmed fish [48]. The antioxidant protection provided by taurine contributes to muscle structure preservation, reducing lipid peroxidation and improving filet quality. Greater resistance to oxidative stress enhances overall fish well-being, promoting more efficient growth. Adding taurine to farmed fish diets is a key strategy to improve their antioxidant capacity and compensate for the limitations of plant-based diet ingredients [48].

### 3.6. Discussion on Key Metabolites and K-Value Assessment

In our study, farmed seabream from Greece exhibited an initial K-value above 20% (Figure 6), potentially due to inadequate handling during transportation. The K-value is closely correlated with sensory properties such as taste, texture, and odor.

Using HR-MAS, we directly quantified IMP, HxR, and Hx without prior extraction, preserving sample integrity and achieving high reproducibility. The K-value enables continuous freshness monitoring during storage and transportation, supports the implementation of quality protocols in the fishing and aquaculture industries, and identifies conditions that extend fish shelf life, minimizing losses due to spoilage.

## 4. Materials and Methods

### 4.1. Chemicals

D_2_O (99.9%) and sodium (3-trimethylsilyl)-2,2,3,3-tetradeuteriopropionate (TSP) were purchased from Sigma–Aldrich (Sigma-Aldrich, St. Louis, MO, USA).

### 4.2. Specimen Collection and Sample Preparation

Three individuals of gilthead seabream were collected from wild populations (via fishing) and aquaculture, with the farmed fish originating from two different locations: Spain and Greece. The fish was purchased from a local fish market in June 2023. The specimens were stored under refrigeration at 4 °C for 18 to 23 days, simulating pre-sale storage conditions. Tissue samples were collected five times from wild-caught seabream and four times from farmed specimens. All samples were preserved at −80 °C for subsequent analysis.

### 4.3. NMR Experiments

Approximately 8–10 mg of white muscle tissue from gilthead seabream was analyzed using HRMAS at 4 °C to minimize tissue deterioration and prevent the degradation of thermolabile compounds. ^1^H-HRMAS NMR spectroscopy was performed at 500.13 MHz on a Bruker AMX500 spectrometer operating at 11.7 T (Bruker, Rheinstetten, Germany).

The samples were loaded into a 50 μL zirconium oxide rotor equipped with a cylindrical insert, along with 20 μL of a 0.1 mM TSP solution in D_2_O. The rotor was spun at 4200 Hz to minimize spinning sideband artifacts while preserving the structural and chemical integrity of the samples during analysis. Under these conditions, no noticeable degradation was observed. The first sample acquisition in each session was repeated two times to verify reproducibility, confirming no differences among spectra due to the high precision of the NMR system.

One-dimensional (1D) solvent-suppressed spectra were acquired using the NOESYPRESAT pulse sequence to suppress water resonance and minimize B_0_ and B_1_ inhomogeneity effects. Each spectrum was acquired with 16k data points, averaged over 128 scans, with a total acquisition time of approximately 7 min. The pulse sequence included a relaxation delay of 2 s, a mixing time (tm) of 150 ms, t1 fixed at 3 μs, and a spectral width of 8333.33 Hz. Spectral data were processed using TOPSPIN software (version 4.2, Bruker, Rheinstetten, Germany). Free induction decays were multiplied by an exponential weighting function corresponding to a line broadening of 0.3 Hz before Fourier transformation. Spectra were manually phased, baseline-corrected and referenced to TSP at δ = 0 ppm [8,9].

Two-dimensional (2D) NMR experiments (COSY-HRMAS) were conducted to assist in the assignment of signals observed in the ^1^H-HRMAS spectra. COSY-HRMAS spectra were acquired with water presaturation during a 1 s relaxation delay, with a spectral width of 8333 Hz in both dimensions, 2 k data points in f2, and 384 increments in f1. An unshifted sinusoidal window function was applied in both dimensions, and zero filling was performed in the f1 dimension. The COSY experiment included zero filling in f1 and unshifted squared sinusoidal window functions prior to Fourier transformation [8,9].

Gradient-selected HSQC-HRMAS experiments were performed using the following parameters: 95 µs for GARP 13C decoupling, with spectral widths of 8333 Hz in the ^1^H dimension and 21 kHz in the ^13^C dimension. The acquisition utilized 2 k data points in the f2 dimension and 256 increments in the f1 dimension. Prior to Fourier transformation, zero filling was applied in the f1 dimension, and an unshifted squared sinusoidal window function was used in both dimensions [8,9].

Control ^1^H-NMR spectra were acquired between consecutive 2D experiments to confirm spectral consistency. No differences were observed in ^1^H-NMR spectra measured from the same sample within 24 h, demonstrating the stability of the experimental conditions.

Standard solvent-suppressed spectra (NOESYPRESAT), COSY-HRMAS, and HSQC-MAS were acquired at the facilities of the Complutense University of Madrid (Centro de Bioimagen Complutense BIOIMAC) following the protocol outlined in the referenced article [8], using identical parameters for all experiments. Two-dimensional (2D) NMR experiments were conducted on the gilthead seabream samples, and the resulting 2D spectra were utilized to aid in the assignment of signals in the ^1^H-HRMAS NMR spectra. All experiments are available in the repository Mendeley Data (see Data Availability Statement section).

The NOESYPRESAT spectra were normalized, and this normalization involved dividing each spectrum intensity by the total intensities of each spectrum, to minimize the effect of different concentrations [11] and reducing them to ASCII files using TopSpin (Bruker, Rheinstetten, Germany) and aligning them using icoshift (version 1.0; available at www.models.kvl.dk (accessed on 4 February 2025) [12] in MATLAB R2022b (Math-Works, Natick, MA, USA). The region of water (4.60–4.95 ppm) and high and low fields (<0.5 ppm and 10 ppm, respectively) were removed. The signals were assigned using The Human Metabolome Database (HMDB, https://hmdb.ca/, accessed on 4 February 2025) and the literature cited in this study. Multivariate data analysis by robust Principal Component Analysis (robPCA) [13] was carried out using the LIBRA toolbox (available at https://wis.kuleuven.be/stat/robust/LIBRAfiles/LIBRA-home-orig (accessed on 4 February 2025)).

The H-HRMAS NMR technique employed in this study primarily facilitates qualitative comparisons among samples. Although it would provide reliable relative quantifications of certain metabolites, precise absolute quantifications were not within the scope of this analysis.

## 5. Conclusions

Gilthead seabream exhibits a series of metabolic trade-offs designed to optimize energy and nutrient utilization in its pelagic environment. Understanding these adaptations is essential for managing wild populations and improving aquaculture practices, ensuring a balance between biological performance, fish quality, health status, and sustainability. Differences in muscle quality and metabolism between wild and farmed seabream reflect specific metabolic adaptations to environment and diet. Elevated levels of metabolites such as creatine, taurine, lactate, and TMAO in wild seabream indicate greater muscle activity and a more diverse microbiota. Conversely, limitations in microbiota diversity and higher lipid storage in farmed seabream highlight the effects of aquaculture practices on metabolism and tissue quality.

The K-value emerges as a reliable indicator of fish freshness, and HR-MAS offers an innovative and efficient methodology for quality control in both wild and farmed seabream. Additionally, further studies exploring gene expression related to lipid metabolism in wild and farmed seabream could complement this analysis, providing deeper insights into the observed lipid profile differences. Recent research has characterized the stress proteome and metabolome in gilthead seabream (*Sparus aurata*), highlighting altered pathways in the liver, a central organ in stress response. These findings could offer valuable insights for future investigations into fish welfare [5].

Study Limitations: This study analyzed only three specimens of each fish type (wild, farmed in Spain, and farmed in Greece), which limits the generalizability of the results. Future studies with larger sample sizes are recommended to validate and expand upon these findings.

## Figures and Tables

**Figure 1 molecules-30-00770-f001:**
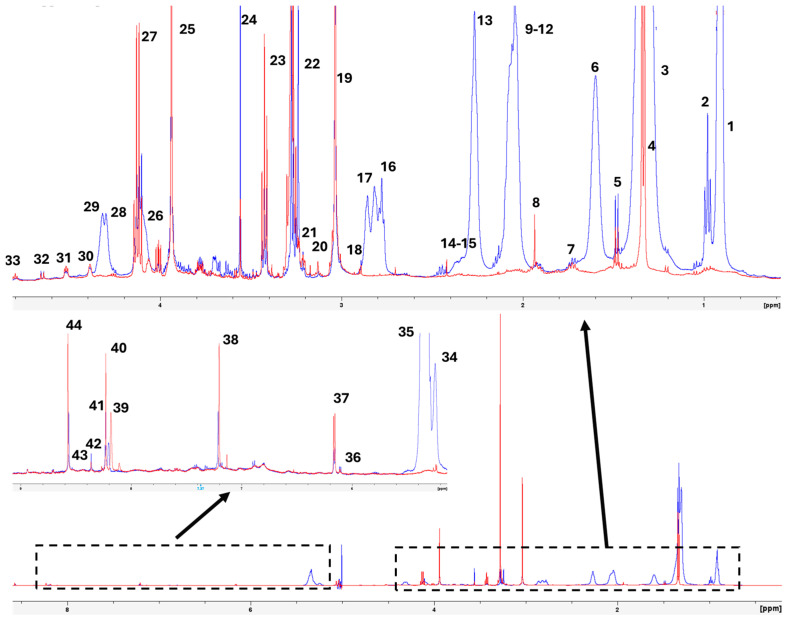
Representative 1D 1H-HRMAS NMR spectrum of muscle tissue from wild gilthead seabream (red line) and farmed gilthead seabream (blue line). The spectra correspond to the initial samples. The boxes present the regions of the spectra that are magnified to see them more clearly. The assignation of the peak indicated by a number is shown in Table 1.

**Figure 2 molecules-30-00770-f002:**
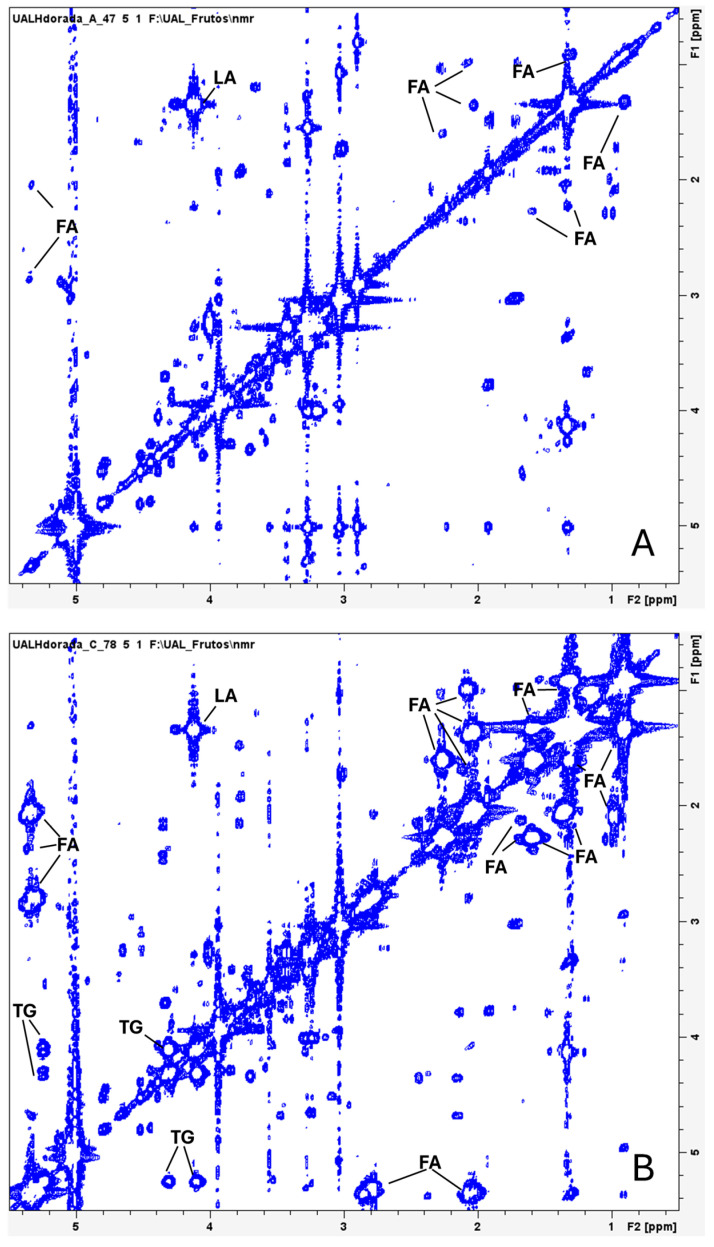
Water-suppressed 1H,1H-COSY HRMAS spectra of muscle tissue from wild gilthead seabream (**A**) and farmed gilthead seabream (**B**). The spectra correspond to the initial samples. Fatty acids (FAs), triglycerides (TGs) and lactate (LA) are indicated in the spectra.

**Figure 3 molecules-30-00770-f003:**
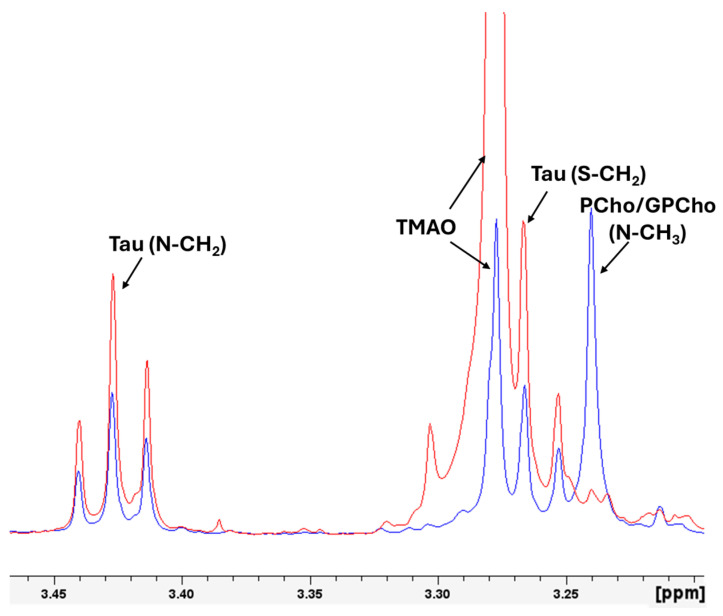
Magnified representation of the region of the 1D 1H-HRMAS NMR spectra of muscle tissue from wild gilthead seabream (red line) and farmed gilthead seabream (blue line), with the signals of the Phosphorylcholine (Pcho)/Glycerolphosphorylcholine (GPCho) (N-CH_3_, 3.24 ppm), taurine (Tau) (S-CH_2_, 3.26), (TMAO (N-CH_3_, 3.28 ppm), taurine (Tau) (N-CH_2_, 3.42). The spectra correspond to the initial samples. The signals of TMAO and Pcho/GPCho are singlets, and the signals of Tau are triplets.

**Figure 4 molecules-30-00770-f004:**
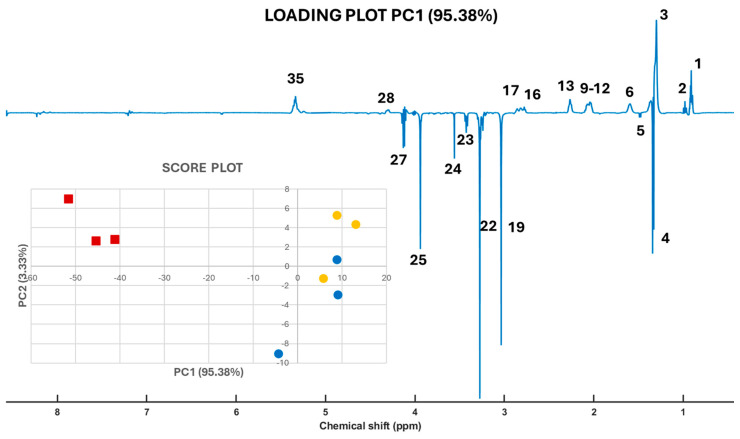
Loadings plot of the 1D pseudo-spectra of PC1 and scores (insert) plots for robPCA of 1D 1H-HRMAS NMR spectra of muscle tissue from wild gilthead seabream (red squares) and farmed gilthead seabream (blue circles from Spain and orange circles from Greece). The assignation of the peak indicated by a number is shown in Table 1.

**Figure 5 molecules-30-00770-f005:**
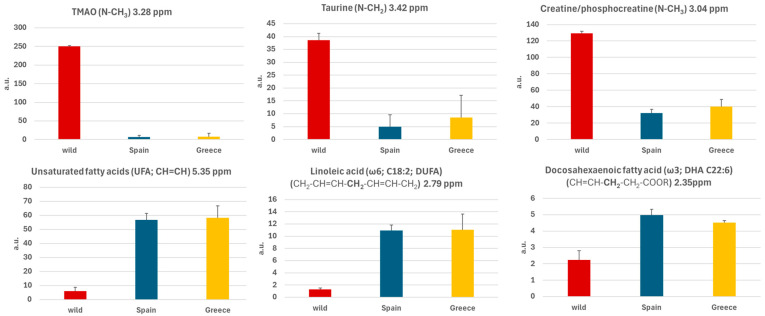
Histograms of the mean values of the integral of the area under the signal of the selected metabolites (TMAO, taurine and creatine/phosphocreatine) and lipids (unsatured fatty acids, linoleic acid and docosahexaenoic acid) and standard errors obtained for samples from wild gilthead seabream and farmed gilthead seabream (from Spain and from Greece).

**Figure 6 molecules-30-00770-f006:**
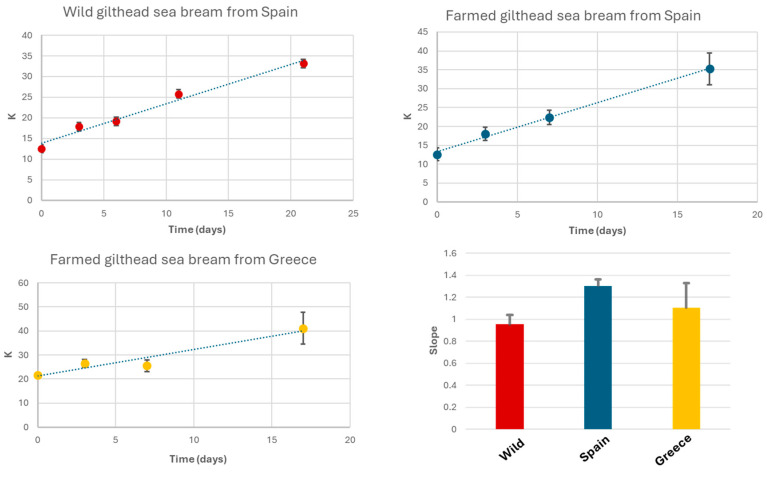
Evolution of K-values in wild gilthead seabream, seabream farmed in Spain, and seabream farmed in Greece, stored at 4 °C in a refrigerator. Each data point represents the mean of measurements from three fish, with vertical bars indicating the standard deviation of these replicates. The insert in the lower right corner represents the slopes of the K-value evolution curves.

**Table 1 molecules-30-00770-t001:** Signal assignments for 1H-HRMAS NMR spectrum from muscle tissue from wild gilthead seabream (*Sparus aurata*).

Peak	Compound	Group	1H (ppm)	Mult.
1	Fatty acids (FAs)	CH_3_	0.91	t
2	Fatty acids (ω-3)	CH=CH-CH_2_-CH_3_	0.98	t
3	Fatty acids (FAs)	(CH_2_)*_n_*	1.30	
4	Lactic acid (LA)	-CH_3_	1.34	d
5	Alanine (Ala)	-CH_3_	1.48	d
6	Fatty acids (FAs)	CH_2_-CH_2_-CH_2_-COOR	1.60	
7	Eicosapentaenoic fatty acid (ω-3; EPA C20:5)	CH=CH-CH_2_-CH_2_-CH_2_-COOR	1.67	
8	Acetic acid (AA)	CH_3_	1.93	s
9	Unsaturated fatty acids (UFAs)	(CH_2_)*_n_*-CH_2_-CH=CH-CH_2_-(CH_2_)*_n_*	2.04	
10	Linoleic acid (ω-6; C18:2; DUFA)	CH_2_-CH_2_-CH=CH-CH_2_-CH=CH-CH_2_-CH_2_	2.08	
11	Unsaturated fatty acids (ω-3)	CH=CH-CH_2_-CH_3_	2.09	
12	Eicosapentaenoic fatty acid (ω-3; EPA C20:5)	ROOC-CH_2_-CH_2_-CH_2_-CH=CH	2.12	
13	Fatty acids (FAs)	CH_2_-CH_2_-COOR	2.27	
14	Docosahexaenoic fatty acid (ω-3; DHA C22:6)	CH=CH-CH_2_-CH_2_-COOR	2.35	
15	Docosahexaenoic fatty acid (ω-3; DHA C22:6)	CH=CH-CH_2_-CH_2_-COOR	2.35	
16	Linoleic acid (ω-6; C18:2; DUFA)	CH_2_-CH=CH-CH_2_-CH=CH-CH_2_	2.78	
17	Polyunsaturated fatty acids (PUFAs)	CH=CH-CH_2_-CH=CH-CH_2_-CH=CH	2.82	
18	Trimethylamine (TMA)	N-CH_3_	2.91	s
19	Creatine/phosphocreatine (Cr/PCr)	N-CH_3_	3.04	s
20	Phosphorylcholine (PCho)/Glycerophosphoryl	N-CH_3_	3.24	
21	Taurine (Tau)	S-CH_2_	3.27	t
22	Trimethylamine oxide (TMAO)	N-CH_3_	3.28	s
23	Taurine (Tau)	N-CH_2_	3.42	t
24	Glycine (Gly)	-CH	3.56	s
25	Creatine/phosphocreatine (Cr/PCr)	-CH_2_	3.94	s
26	Triglycerides (TGs)	CH_2_-α(Gly)	4.12	
27	Lactic acid (La)	α-CH	4.12	q
28	Triglycerides (TGs)	CH_2_-α’(Gly)	4.30	
29	Glycerophosphorylcholine (GPCho)	α-CH_2_	4.32	
30	Inosine (Ino)	CH-3 (Rib)	4.45	t
31	Anserine (Ans)	α-CH (His)	4.52	m
32	β-Glucose (β-Glc)	CH-1	4.65	d
33	Inosine (Ino)	CH-2 (Rib)	4.80	t
34	Triglycerides (TGs)	CH (Gly)	5.25	
35	Unsaturated fatty acids (UFAs)	CH=CH	5.35	
36	Hypoxanthine ribose (HxR)	CH-1 (Rib)	6.11	d
37	Inosine 5′-phosphate (IMP)	CH-1 (Rib)	6.16	d
38	Anserine (Ans)	CH-5 (His)	7.20	s
39	Hypoxanthine (Hx)	CH-8	8.19	s
40	Inosine (Ino)	CH-8 (Purin)	8.23	s
41	Inosine 5′-phosphate (IMP)	CH-8	8.24	s
42	Formic acid (FA)	CH	8.36	s
43	Adenosine tri/di/monophosphate (ATP/ADP)	CH-2	8.49	s
44	Anserine (Ans)	CH-2 (His)	8.57	s

s, singlet; d, doublet; t, triplet; q, quartet; m, multiplet.

## Data Availability

The data presented in this study are openly available in Marhuenda, Frutos (2024), “HR-MAS *Sparus aurata*”, Mendeley Data, V2, https://data.mendeley.com/drafts/rjmjj5vwf2 (accessed 4 February 2025).

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
