# Peer review of "Metabolomic Insights into Wild and Farmed Gilthead Seabream (*Sparus aurata*): Lipid Composition, Freshness Indicators, and Environmental Adaptations"

_molecules, 2025, doi:10.3390/molecules30040770_

Round 1

Reviewer 1 Report

Comments and Suggestions for Authors

Avoid duplicating in the keyword list words already included in the title.

Introduction is perhaps too long; try to reduce it by 20% more or less.

The manuscript should be written in past tense as the experiments are already finished.

Limitations: Using only 3 specimens of each type of fish is a limiting factor for the results generated. This point should be clearly stated in the manuscript text.

Data provided in lines 184-194 are results of the study; thus, they have to be presented in the R&D section and not in the M&M section.

Does the analytical method used (H-HRMAS NMR) allow for an estimation of the contents of the compounds under study? If not, and only qualitative comparisons among samples can be done; this information should be clearly stated in the manuscript.

Where the standards described in the manuscript used for the correct identification of compounds listed in Figure 1? This information should be clearly stated in the manuscript.

I do not see any table of figure with the concentration of the fatty acids. Besides, the discussion and conclusions include comments about levels of creatine, taurine, lactate, and TMAO but I do not see these values anywhere.

Author Response

Comment 1: Avoid duplicating in the keyword list words already included in the title.

Response 1: Done.

Comment 2: Introduction is perhaps too long; try to reduce it by 20% more or less.

Response 2: Done.

Comment 3: The manuscript should be written in past tense as the experiments are already finished.

Response 3: Done.

Comment 4: Limitations: Using only 3 specimens of each type of fish is a limiting factor for the results generated. This point should be clearly stated in the manuscript text.

Response 4: A section addressing the study's limitations has been added at the end of the manuscript (lines 499-501).

Comment 5: Data provided in lines 184-194 are results of the study; thus, they have to be presented in the R&D section and not in the M&M section.

Response 5: This paragraph has been moved to the Results and Discussion (R&D) section as suggested. It is indeed more appropriate there (lines 214-224).

Comment 6: Does the analytical method used (H-HRMAS NMR) allow for an estimation of the contents of the compounds under study? If not, and only qualitative comparisons among samples can be done; this information should be clearly stated in the manuscript.

Response 6: A paragraph clarifying this point has been added to the Materials and Methods (M&M) section (lines 207-210).

Comment 7: Where the standards described in the manuscript used for the correct identification of compounds listed in Figure 1? This information should be clearly stated in the manuscript.

Response 7: This information is detailed in the R&D section (lines 204-206). Spectra from compounds available in databases, such as the Human Metabolome Database (HMDB; https://hmdb.ca/), and published studies were used for accurate identification.

Comment 8: I do not see any table of figure with the concentration of the fatty acids. Besides, the discussion and conclusions include comments about levels of creatine, taurine, lactate, and TMAO but I do not see these values anywhere.

Response 8: As suggested, a histogram has been included with data on fatty acids and other metabolites, allowing visualization of the differences between wild and farmed seabream (lines 279-282). 

Reviewer 2 Report

Comments and Suggestions for Authors

Review of the paper “Metabolomic Insights into Wild and Farmed Gilthead Seabream (Sparus aurata): Lipid Composition, Freshness Indicators, and Environmental Adaptations”

Frutos C. Marhuenda-Egea,and Pablo Sanchez-Jerez

The study report on HR-MAS NMR analysis of intact fish biopsies to compare wild and farmed seabream from two different countries, in terms of environmental conditions, feeding regimes and freshness index.

The paper is relatively well written. The introduction is complete and the purpose of the study is well defined. There are some repetitions e.g .about HR-MAS line 105-132 and 150-159, or in discussion section, that should be avoided, or regrouped in the same paragraph.

There are some errors that shouldn’t be there, see below, and which suggest that the authors have copied text from another publication without checking it. There are also some imprecisions and errors which need to be corrected.

Methods

- Line 177: 3 individuals of each population have been used for the study. Wasn’t it possible to take 2 samples in each fish, to have a technical replicate?

- Line 184-185: can the difference in average weight between wild and farmed fishes be due to a difference in age of the fishes?

How was calculated weight difference line 194? strictly, the weight difference formulae is (W1 – W2)/mean value of weight, i.e. (W1+W2)/2. So around 56% between wild and farmed fish.

The number indicated: 42.5% and 43.5%, are not weight differences but weight decrease, in percentage, of wild fishes compared to farmed fishes (i.e. (Wwild – Wfarmed)/Wfarmed).

- Line 195: “following classification”: what do the authors mean?

- Line 198: “white muscle tissue from smoked Atlantic salmon” : mistake???

- Line 204: which spinning rate was used?

- Line 208-209: Even if citing a reference paper for NMR experimental parameters, a minimum of important parameters should be provided.

Why didn’t the authors use a T2 filter pulse sequence (CPMG), like in Heude et al. (ref 7), to limit macromolecular signal and enhance small metabolite signal? A diffusion-weighted acquisition could further been acquired to enhance broad lipidic signal and remove sharp metabolite signals.

- Line 220: HSQC experiment is not a homonuclear correlation experiment, but a heteronuclear one (13C-1H).

- Legend figure 1

Line 230: “…1H CPMG HR-MAS ….”:???? The authors have indicated that they have used a NOESYPRESAT pulse sequence like in ref 9 (Castejon et al.), and not a CPMG one…

The assignment of peaks to metabolites or lipids should be provided in a table, supplemental or not. Concerning lipid resonances, the assignment proposed in the legend is not correct. The lipid peaks observed by NMR correspond to a position of methyl, methylene or methyne groups along fatty acid chains, whatever the length of the chain. This is correctly indicated in the table 1 of Castelon et al. This mean that proton NMR does not allow to identify individual lipids, but 13C NMR could. For example, the peak 3 corresponds to (CH2)n along fatty acids. The 8 one correspond to all methylene groups like the one in bold (CH2-CH2-COOR), and not only to “Docosahexaenoic Fatty Acid (ω-3; EPA C22:6). Etc.

Or, if the authors have arguments to assign lipid resonances to specific lipids, they must provide them precisely, and provide the chemical shift and position of the proton group along the fatty acid chain.

Figure 1 is of insufficient quality. In ref 9 the authors have a much better spectral resolution. The authors could increase the amplitude of the spectra and truncate the intense singlets.

How were calculated the concentrations of metabolites for calculation of K score? This is not obvious with HR-MAS NMR, which gives access to a number of mole, and not to a concentration like in liquid NMR. In Heude et al. they used a sucrose solution with a known number of mole to calculate the concentrations. In that case the exact mass of each sample must been known.

Additionally, line 357-358, the authors say that “wild gilthead seabream exhibit significantly higher levels of creatine, taurine, and TMAO”: I haven’t seen any statistical test that could allow to claim that the difference is significant. Moreover, since the weight of samples included into the rotors can be different between samples, the strict observation of spectra is more complex than with solutions, and is not direct.

The results concerning other metabolites than those included into the k score, and concerning lipid composition, must be presented in result section and not in discussion one.

Concerning lipids, the comparison between wild and farmed fish is also qualitative, and does not take into account for the difference in weight of the sample put into the rotor.

Despite that, it is clear that there is much less lipids and macromolecules in wild fish.

Is there any difference between Greek and Spanish farm?

Discussion

See above for a reorganisation of discussion with some results needing to be displaced to result section.

The discussion concerning taurine also need to be re-organized, and arguments grouped in the same paragraph.

Concerning difference in lipids and metabolites between wild and farmed fishes: it would be interesting that the authors comment on the difference in average weight, which is very important.

Author Response

Review of the paper “Metabolomic Insights into Wild and Farmed Gilthead Seabream (Sparus aurata): Lipid Composition, Freshness Indicators, and Environmental Adaptations”

Frutos C. Marhuenda-Egea,and Pablo Sanchez-Jerez

The study report on HR-MAS NMR analysis of intact fish biopsies to compare wild and farmed seabream from two different countries, in terms of environmental conditions, feeding regimes and freshness index.

Comment 1: The paper is relatively well written. The introduction is complete and the purpose of the study is well defined. There are some repetitions e.g .about HR-MAS line 105-132 and 150-159, or in discussion section, that should be avoided, or regrouped in the same paragraph.

Response 1: Done. This section of the introduction has been rewritten.

There are some errors that shouldn’t be there, see below, and which suggest that the authors have copied text from another publication without checking it. There are also some imprecisions and errors which need to be corrected.

Methods

Comment 2: - Line 177: 3 individuals of each population have been used for the study. Wasn’t it possible to take 2 samples in each fish, to have a technical replicate?

Response 2: As indicated, we initially took two samples per fish. However, the differences observed in the spectra were minimal. Given the time and cost associated with analyzing double the samples, we concluded that additional replicates were not justified by the marginal information gained.

Comment 3: - Line 184-185: can the difference in average weight between wild and farmed fishes be due to a difference in age of the fishes?

Response 3: This is a possibility we considered. Farmed fish are harvested under strict control to achieve a specific size, whereas wild fish are caught without this consideration, as long as they meet minimum size regulations. For our study, we aimed to collect fish of similar size (length), although weight differences were significant. A wild fish matching the weight of a farmed fish would likely be much older.

A key factor in animal breeding is to achieve a certain size that allows its commercialization in an adequate time, which is usually the shortest possible. Always within the animal welfare considerations determined by the corresponding regulations (lines 317-329).

Comment 4: How was calculated weight difference line 194? strictly, the weight difference formulae is (W1 – W2)/mean value of weight, i.e. (W1+W2)/2. So around 56% between wild and farmed fish.

The number indicated: 42.5% and 43.5%, are not weight differences but weight decrease, in percentage, of wild fishes compared to farmed fishes (i.e. (Wwild – Wfarmed)/Wfarmed).

Response 1: Weight differences were calculated as suggested by the reviewer. These calculations have been specified in the corresponding paragraph (lines 225-227).

Comment 5: - Line 195: “following classification”: what do the authors mean?

Response 1: This phrase was unclear and has been removed.

Comment 6: - Line 198: “white muscle tissue from smoked Atlantic salmon” : mistake???

Response 1: Corrected.

Comment 7: - Line 204: which spinning rate was used?

Response 1: We have expanded the description of the experiments (lines 141-210). The spinning rate used was 4200 Hz, optimized to prevent potential sample alterations during the experiment.

Comment 8: - Line 208-209: Even if citing a reference paper for NMR experimental parameters, a minimum of important parameters should be provided.

Why didn’t the authors use a T2 filter pulse sequence (CPMG), like in Heude et al. (ref 7), to limit macromolecular signal and enhance small metabolite signal? A diffusion-weighted acquisition could further been acquired to enhance broad lipidic signal and remove sharp metabolite signals.

Response 1: The description of the experiments has been expanded as suggested (lines 141-210). Although we could have chosen the CPMG experiment, we had not yet optimized it for this type of sample. Therefore, we opted for the NOESYPRESAT sequence, which still provided valuable information. For future experiments, we are optimizing the CPMG experiment, as it could yield highly informative results.

Comment 9: - Line 220: HSQC experiment is not a homonuclear correlation experiment, but a heteronuclear one (13C-1H).

Response 1: Corrected.

Comment 10: - Legend figure 1

Line 230: “…1H CPMG HR-MAS ….”:???? The authors have indicated that they have used a NOESYPRESAT pulse sequence like in ref 9 (Castejon et al.), and not a CPMG one…

Response 1: The error has been corrected. Indeed, we conducted the analyses using the NOESYPRESAT pulse sequence.

Comment 11: The assignment of peaks to metabolites or lipids should be provided in a table, supplemental or not. Concerning lipid resonances, the assignment proposed in the legend is not correct. The lipid peaks observed by NMR correspond to a position of methyl, methylene or methyne groups along fatty acid chains, whatever the length of the chain. This is correctly indicated in the table 1 of Castelon et al. This mean that proton NMR does not allow to identify individual lipids, but 13C NMR could. For example, the peak 3 corresponds to (CH2)n along fatty acids. The 8 one correspond to all methylene groups like the one in bold (CH2-CH2-COOR), and not only to “Docosahexaenoic Fatty Acid (ω-3; EPA C22:6). Etc.

Or, if the authors have arguments to assign lipid resonances to specific lipids, they must provide them precisely, and provide the chemical shift and position of the proton group along the fatty acid chain.

Response 1: The peak assignments have been presented in table format (Table 1), including all relevant data suggested by the reviewer (line 276).

Comment 12: Figure 1 is of insufficient quality. In ref 9 the authors have a much better spectral resolution. The authors could increase the amplitude of the spectra and truncate the intense singlets.

Response 1: Figure 1 has been redone to improve its graphical quality.

Comment 13: How were calculated the concentrations of metabolites for calculation of K score? This is not obvious with HR-MAS NMR, which gives access to a number of mole, and not to a concentration like in liquid NMR. In Heude et al. they used a sucrose solution with a known number of mole to calculate the concentrations. In that case the exact mass of each sample must been known.

Response 1: The K value is a percentage calculated from the integral of the signals specified in the formula. Since it is a percentage, it does not require the quantitative determination of compound concentrations. The signal integrals in NMR spectra are considered directly proportional to the concentrations of the compounds generating the signal (line 287).

Comment 14: Additionally, line 357-358, the authors say that “wild gilthead seabream exhibit significantly higher levels of creatine, taurine, and TMAO”: I haven’t seen any statistical test that could allow to claim that the difference is significant. Moreover, since the weight of samples included into the rotors can be different between samples, the strict observation of spectra is more complex than with solutions, and is not direct.

Response 1: A new figure (Figure 3) has been included, showing histograms of the selected signal areas. These represent the mean values for each fish type (wild, farmed in Spain, and farmed in Greece), along with calculated standard errors. The signal areas in the spectra were determined from normalized spectra, ensuring accurate comparisons (line 279).

Comment 15: The results concerning other metabolites than those included into the k score, and concerning lipid composition, must be presented in result section and not in discussion one.

Response 1: Done (lines 284-307).

Comment 16: Concerning lipids, the comparison between wild and farmed fish is also qualitative, and does not take into account for the difference in weight of the sample put into the rotor.

Despite that, it is clear that there is much less lipids and macromolecules in wild fish.

Response 1: Indeed, as shown in the spectra (Figures 1 and 2), the differences between samples used in the rotor are minimal and do not justify the much higher lipid content observed in farmed seabream compared to wild seabream.

Figure 1 illustrates the significant differences between wild and farmed seabream. Figure 2 further highlights these differences while enabling precise signal assignments and sample comparisons.

Figure 3 presents histograms of specific molecules (polar metabolites and lipids). Wild seabream exhibit higher levels of polar metabolites such as TMAO, taurine, creatine, and phosphocreatine, likely associated with their higher muscle mass. In contrast, farmed seabream display higher levels of lipids (UFA, DUFA, and DHA) in muscle tissue, influenced by their diet. One notable lipid is linoleic acid, present in aquaculture feeds.

Comment 17: Is there any difference between Greek and Spanish farm?

Response 1: Figure 3 shows minimal differences between seabream farmed in Spain and Greece. This is likely due to the use of similar aquaculture feed for both groups (lines 323-329).

Discussion

Comment 18: See above for a reorganisation of discussion with some results needing to be displaced to result section.

The discussion concerning taurine also need to be re-organized, and arguments grouped in the same paragraph.

Response 1: The discussion has been reorganized as suggested by the reviewer.

Comment 19: Concerning difference in lipids and metabolites between wild and farmed fishes: it would be interesting that the authors comment on the difference in average weight, which is very important.

Response 19: Thank you for your observation. We have included additional details in the manuscript regarding the morphological characteristics of the fish, specifically addressing differences in average weight and size. Wild gilthead seabream had an average weight of 288.3 ± 35.9 g, whereas farmed seabream from Spanish aquaculture weighed 510.4 ± 95.6 g, and those farmed in Greece weighed 501.0 ± 23.4 g. The total length (TL) was 28.0 ± 1.3 cm for wild seabream, 30.1 ± 1.6 cm for Spanish farmed seabream, and 29.7 ± 0.3 cm for Greek farmed seabream. Similarly, the fork length (FL) was 26.0 ± 0.9 cm, 28.5 ± 0.9 cm, and 28.0 ± 0.5 cm, respectively. This explanation has been incorporated into the Results section to provide further context (lines 214-227).

The weight difference between wild and farmed seabream ranged from 42.5% to 43.5%, while the length difference ranged from 7% to 8%. These differences likely result from the controlled feeding and farming conditions in aquaculture, which are optimized to achieve commercial sizes more efficiently. These differences in morphology, particularly weight, may influence the observed lipid and metabolite profiles, as farmed seabream exhibit higher lipid content due to their diet, whereas wild seabream show a greater proportion of polar metabolites associated with muscle activity (lines 318-329). 

Reviewer 3 Report

Comments and Suggestions for Authors

The paper entitled "Metabolomic Insights into Wild and Farmed Gilthead Seabream (Sparus aurata): Lipid Composition, Freshness Indicators, and Environmental Adaptations" by Marhuenda-Egea and Sanchez-Jerez described an interesting application of HRMAS-NMR in food science, namely for the distinction between farmed and wild seabream.

The HRMAS-NMR is well known and fully accepted by the academic community, so the approach used in this paper has a solid scientific base.

I suggest to improve the quality of Figure 1, usually the quality of peaks is higher, to add (eventually in the supplementary material) a 2D spectrum used for the assignments.

Line 198 "white muscle tissue from smoked Atlantic salmon", probably it is not salmon.

Author Response

The paper entitled "Metabolomic Insights into Wild and Farmed Gilthead Seabream (Sparus aurata): Lipid Composition, Freshness Indicators, and Environmental Adaptations" by Marhuenda-Egea and Sanchez-Jerez described an interesting application of HRMAS-NMR in food science, namely for the distinction between farmed and wild seabream.

The HRMAS-NMR is well known and fully accepted by the academic community, so the approach used in this paper has a solid scientific base.

Comment 1: I suggest to improve the quality of Figure 1, usually the quality of peaks is higher, to add (eventually in the supplementary material) a 2D spectrum used for the assignments.

Response 1: Done. We have modified Figure 1 and included ¹H,¹H-COSY HRMAS spectra (Figure 2).

Comment 2: Line 198 "white muscle tissue from smoked Atlantic salmon", probably it is not salmon.

Response 1: Corrected.

Round 2

Reviewer 1 Report

Comments and Suggestions for Authors

Authors have incorporated all my comments/suggestions in the revised version of their manuscript. 

Author Response

Many thanks for your contributions, which have greatly helped us to improve the original manuscript.

Reviewer 2 Report

Comments and Suggestions for Authors

Second review of the paper “Metabolomic Insights into Wild and Farmed Gilthead Seabream (Sparus aurata): Lipid Composition, Freshness Indicators, and Environmental Adaptations”

Frutos C. Marhuenda-Egea,and Pablo Sanchez-Jerez

The paper has been notably improved, however there are still some issues and mistakes, mainly:

- there are lot of repetition of the same arguments,

- the authors claim Line 131 that their aim was to characterize muscle and liver metabolome in the study, and I didn’t see any results about liver …

- there is a discrepancy between the spectral assignment and the peaks used to calculate the K score.

In detail:

Methods

Line 159-161: already said line 153-154

Line 162-163: already said line 154-155

Line 201: “…the spectra were normalized…”: which normalization was used? This is a major point, mainly for the histograms of figure 3. The authors say in legend figure 3 that this is the area under the peaks. However, the weight of the sample directly influences signal amplitude in HR-MAS NMR (unlike liquid NMR, in which concentration of molecule, in gram per liter or mole per liter, influence the amplitude of peaks). So for a given metabolite, the simple amplitude of peak are not comparable between spectra. Except if normalisation is applied. However, I agree that for the K value, as this is a ratio of different integrals inside each spectrum, this does not require quantitative determination.

Identically, the authors say, in their response to my comment 13: “”the signal integrals in NMR spectra are considered directly proportional to the concentration of the compounds generating the signal”. This is true in liquid NMR, and not in HR-MAS NMR. This is due to the geometry of the HR-MAS probehead.

Results:

Line 223-227: It is not necessary to give the formulae, because everybody know how is calculated a decrease. The authors must simply be precise and clear, e.g. “the weight decrease in wild seabream compared to farmed ones was….”. So the term “weight difference” has not to be used. However, it only makes sense to calculate weight decrease if it supports any hypothesis. This does not seem relevant in this study.

Line 240-241: already said line 231-233.

Line 292 to 294: “These signals include the doublet of IMP at 6.14 ppm (CH-1 Ribose), the doublet of HxR 292 at 6.09 ppm (CH-1 Ribose), the singlet of Hx at 8.18 ppm (CH-8), and the singlet of 293 ATP/ADP/AMP at 8.49 ppm (CH-2 Purine)”. None of these chemical shifts are found table 1!. This is not acceptable.

Line 250: concerning TMAO: in the spectral region where it appears, there is also a triplet of taurine, which can overlap with TMAO, depending on spectral resolution. How did the authors take this into account? A zoom on the spectral region of TMAO should be provided.

Discussion

Line 318: Again here, the authors didn’t provide a weight difference, but a weight variation in wild fish compared to farmed one.

Line 337: “TMAO can be detected with greater precision using HR-MAS.” Yes HR-MAS NMR avoid extraction, but TMAO detection also depends on spectral resolution, that increases with magnetic field.

Line 331-349: this paragraph should be revised: A lot of repetition of arguments are found at different places of the paragraph, and the last sentence should be displaced. Moreover, the fact that multivariate statistical analysis are feasible with HR-MAS data has been largely demonstrated in the past in several research domain.

Line 421-460: the different arguments and hypothesis about taurine should be brought together in the same place.

Conclusion

Line 484-488: here also, these two sentences are equivalent..

Author Response

Second response to Reviewer 2 Comments

Second review of the paper “Metabolomic Insights into Wild and Farmed Gilthead Seabream (Sparus aurata): Lipid Composition, Freshness Indicators, and Environmental Adaptations”

Frutos C. Marhuenda-Egea,and Pablo Sanchez-Jerez

The paper has been notably improved, however there are still some issues and mistakes, mainly:

- there are lot of repetition of the same arguments,

- the authors claim Line 131 that their aim was to characterize muscle and liver metabolome in the study, and I didn’t see any results about liver …

- there is a discrepancy between the spectral assignment and the peaks used to calculate the K score.

In detail:

Methods

Line 159-161: already said line 153-154

Corrected

Line 162-163: already said line 154-155

Corrected

Line 201: “…the spectra were normalized…”: which normalization was used? This is a major point, mainly for the histograms of figure 3. The authors say in legend figure 3 that this is the area under the peaks. However, the weight of the sample directly influences signal amplitude in HR-MAS NMR (unlike liquid NMR, in which concentration of molecule, in gram per liter or mole per liter, influence the amplitude of peaks). So for a given metabolite, the simple amplitude of peak are not comparable between spectra. Except if normalisation is applied. However, I agree that for the K value, as this is a ratio of different integrals inside each spectrum, this does not require quantitative determination.

The spectra were normalized relative to the total spectrum intensity in order to minimize the effect of different concentrations [11] (lines 198-199). It is important to indicate the method of normalization, as there are different strategies for normalizing NMR spectra.

  1. Bruzzone, C., M. Bizkarguenaga, R. Gil-Redondo, T. Diercks, E. Arana, A. G. de Vicuna, M. Seco, A. Bosch, A. Palazon, I. San Juan, et al. "Sars-cov-2 infection dysregulates the metabolomic and lipidomic profiles of serum." Iscience 23 (2020): 10.1016/j.isci.2020.101645. <Go to ISI>://WOS:000581985500097.

Identically, the authors say, in their response to my comment 13: “”the signal integrals in NMR spectra are considered directly proportional to the concentration of the compounds generating the signal”. This is true in liquid NMR, and not in HR-MAS NMR. This is due to the geometry of the HR-MAS probehead.

While HR-MAS NMR signal intensity remains proportional to proton quantity, as in classical ¹H NMR, various factors such as relaxation effects, sample heterogeneity, and experimental conditions can impact the proportionality. Proper optimization of experimental parameters and sample preparation is essential to ensure quantitative accuracy. When carefully controlled, HR-MAS NMR is a powerful technique for both qualitative and quantitative analysis, particularly in complex systems such as biological tissues or polymers.

To ensure that the signal intensity on HR-MAS NMR accurately reflects the number of protons, several experimental strategies should be employed: adjusting the repetition time (?r) to at least five times the longest ?1 of the sample to allow complete relaxation of all protons between acquisitions; measuring T1 and T2 relaxation times to account for dynamic differences between regions, which can guide parameter optimization for quantitative experiments; and minimizing experimental artifacts by ensuring uniform sample preparation, proper hydration, and optimized centrifugation conditions to improve resolution and reduce spectral distortions. We have tried to take all these factors into account to ensure that the spectra are of sufficient quality for scientific work.

Results:

Line 223-227: It is not necessary to give the formulae, because everybody know how is calculated a decrease. The authors must simply be precise and clear, e.g. “the weight decrease in wild seabream compared to farmed ones was….”. So the term “weight difference” has not to be used. However, it only makes sense to calculate weight decrease if it supports any hypothesis. This does not seem relevant in this study.

Correted (lines 221-223).

Line 240-241: already said line 231-233.

Lines 231 to 233 have been deleted.

Line 292 to 294: “These signals include the doublet of IMP at 6.14 ppm (CH-1 Ribose), the doublet of HxR 292 at 6.09 ppm (CH-1 Ribose), the singlet of Hx at 8.18 ppm (CH-8), and the singlet of 293 ATP/ADP/AMP at 8.49 ppm (CH-2 Purine)”. None of these chemical shifts are found table 1!. This is not acceptable.

Table 1 and Figure 1 were corrected in order to include these signals.

Line 250: concerning TMAO: in the spectral region where it appears, there is also a triplet of taurine, which can overlap with TMAO, depending on spectral resolution. How did the authors take this into account? A zoom on the spectral region of TMAO should be provided.

As suggested by the reviewer, we have included a new figure (Figure 3) that magnifies the region between 3 and 3.5 ppm. This region contains the taurine triplets (S-CH2 at 3.27 ppm, and N-CH2 at 3.42 ppm) and the TMAO singlet (N-CH3 at 3.28 ppm). One of the side signals of the taurine triplet (S-CH2) coincides with the TMAO singlet. We have considered that the contribution of this taurine signal to the TMAO signal is similar for both wild and farmed gilthead seabream, so we have not taken it into account.

Discussion

Line 318: Again here, the authors didn’t provide a weight difference, but a weight variation in wild fish compared to farmed one.

Corrected (line 325).

Line 337: “TMAO can be detected with greater precision using HR-MAS.” Yes HR-MAS NMR avoid extraction, but TMAO detection also depends on spectral resolution, that increases with magnetic field.

Indeed, the resolution increases with the intensity of the magnetic field, but we do not intend to make a quantitative determination of the compound, for which we would have had to add a suitable standard, but rather to be able to compare the metabolites present in the different fish, as well as a correct identification of the signals. This identification is achieved with both 1D and 2D spectra.

Line 331-349: this paragraph should be revised: A lot of repetition of arguments are found at different places of the paragraph, and the last sentence should be displaced. Moreover, the fact that multivariate statistical analysis are feasible with HR-MAS data has been largely demonstrated in the past in several research domain.

This section has been rewritten (lines 337-351).

Line 421-460: the different arguments and hypothesis about taurine should be brought together in the same place.

Corrected.

Conclusion

Line 484-488: here also, these two sentences are equivalent

Corrected.

Round 3

Reviewer 2 Report

Comments and Suggestions for Authors

Third review of the paper “Metabolomic Insights into Wild and Farmed Gilthead Seabream (Sparus aurata): Lipid Composition, Freshness Indicators, and Environmental Adaptations”

Frutos C. Marhuenda-Egea,and Pablo Sanchez-Jerez

The paper has been notably improved, however there are still some issues and mistakes, mainly:

1/ The authors claim Line 131 that “their aim was to characterize muscle and liver metabolome in the study”, and I didn’t see any results about liver …

I already reported that in my second review but this hasn’t been corrected.

2/ Normalization: The authors have added the sentence The spectra were normalized relative to the total spectrum intensity in order to minimize the effect of different concentrations [11] (lines 198-199). “

Yes, I agree that normalization to sum is a robust method, this means: metabolites (amplitude of peaks) are normalized to the total spectrum. I do not understand “spectra normalized to total spectrum intensity”. This must be clarified.

3/ TMAO:

a/ Yes, I agree that the left peak of the S-CH2 triplet of taurine overlap with TMAO, as seen in figure 3.

However, it is seen in figure 4 that taurine (quantified through the 3.42 ppm triplet) is largely higher in wild fish compared to farmed one. So, how can the authors say that “…the contribution of this taurine signal to the TMAO signal is similar for both wild and farmed gilthead seabream, so we have not taken it into account.”

This is false

What is true is that the peak at 3.42 ppm should be identical to the peak at 3.27 (same profile, multiplicity and amplitude). So it may be possible to subtract the contrition of the left peak of taurine at 3.42 to the TMAO peak...

Or: if the authors do not take that into account, they must indicate that their TMAO is not pure but partially overlapped with taurine.

b/ The taurine peak seems equivalent between wild and farmed fishes in figure 3, because the blue spectra (farmed fish) has been amplified, so as taurine peak at 3.42 seems equivalent. But this does not represent the reality (see a/)

This must be indicated in legend figure 3 (e.g. “intensity of each spectrum has been differently amplified for a better visualization of peaks”).

4/ I said in my last review that “ the fact that multivariate statistical analysis are feasible with HR-MAS data has been largely demonstrated in the past in several research domain..”

In fact, I do not understand why do the authors report on the possibility to use PCA and OPLS-DA with HR-MAS data. They didn’t use these statistics, and this does not demonstrate anything about HR-MAS NMR.

Why didn’t they use these multivariate statistics?

Author Response

Third review of the paper “Metabolomic Insights into Wild and Farmed Gilthead Seabream (Sparus aurata): Lipid Composition, Freshness Indicators, and Environmental Adaptations”

Frutos C. Marhuenda-Egea,and Pablo Sanchez-Jerez

The paper has been notably improved, however there are still some issues and mistakes, mainly:

1/ The authors claim Line 131 that “their aim was to characterize muscle and liver metabolome in the study”, and I didn’t see any results about liver …

Corrected.

I already reported that in my second review but this hasn’t been corrected.

2/ Normalization: The authors have added the sentence “The spectra were normalized relative to the total spectrum intensity in order to minimize the effect of different concentrations [11] (lines 198-199). “

Yes, I agree that normalization to sum is a robust method, this means: metabolites (amplitude of peaks) are normalized to the total spectrum. I do not understand “spectra normalized to total spectrum intensity”. This must be clarified.

Normalization consisted of dividing each spectrum intensity by the total intensities of each spectrum, to minimize the effect of different concentrations (lines 198-200).

3/ TMAO:

a/ Yes, I agree that the left peak of the S-CH2 triplet of taurine overlap with TMAO, as seen in figure 3.

However, it is seen in figure 4 that taurine (quantified through the 3.42 ppm triplet) is largely higher in wild fish compared to farmed one. So, how can the authors say that “…the contribution of this taurine signal to the TMAO signal is similar for both wild and farmed gilthead seabream, so we have not taken it into account.”

This is false

What is true is that the peak at 3.42 ppm should be identical to the peak at 3.27 (same profile, multiplicity and amplitude). So it may be possible to subtract the contrition of the left peak of taurine at 3.42 to the TMAO peak...

Or: if the authors do not take that into account, they must indicate that their TMAO is not pure but partially overlapped with taurine.

Figures 3 and 5 have been corrected:

The value of the integral of the TMAO (N-CH3) signal at 3.28 ppm was corrected by subtracting the value of the integral of the left peak of the taurine triplet (S-CH2) at 3.27 ppm, for each sample (lines 265-267).

b/ The taurine peak seems equivalent between wild and farmed fishes in figure 3, because the blue spectra (farmed fish) has been amplified, so as taurine peak at 3.42 seems equivalent. But this does not represent the reality (see a/)

This must be indicated in legend figure 3 (e.g. “intensity of each spectrum has been differently amplified for a better visualization of peaks”).

There was a confusion with the colors of the spectra in figure 3 and this has been corrected.

4/ I said in my last review that “ the fact that multivariate statistical analysis are feasible with HR-MAS data has been largely demonstrated in the past in several research domain..”

In fact, I do not understand why do the authors report on the possibility to use PCA and OPLS-DA with HR-MAS data. They didn’t use these statistics, and this does not demonstrate anything about HR-MAS NMR.

Why didn’t they use these multivariate statistics?

We have included principal component analysis (robust PCA) which allows us to confirm that lipid signals and other metabolites (TMAO, taurine or cretin) are the key to differentiate between wild and farmed sea bream (Figure 4):

Multivariate data analysis by robust Principal Component Analysis (robPCA) [13] was carried out using the LIBRA toolbox (available at ww.wis.kuleuven.ac.be/stat/robust.html). (Lines 205-207).

We employed robust Principal Component Analysis (robPCA) [13] to identify the key molecular signals that distinguish gilthead seabream based on their origin (wild or farmed). The pseudo-spectrum signals, corresponding to the loadings of Principal Component 1 (PC1, 95.38%) (Figure 4), indicate that positive signals are more intense in the 1D 1H-HRMAS NMR spectra of farmed fish samples from both Spain and Greece. Conversely, negative signals in the pseudo-spectrum suggest greater intensity in the 1D 1H-HRMAS NMR spectra of wild seabream samples. The loadings presented in Figure 4 were instrumental in differentiating between wild and farmed seabream, as illustrated in the insert of Figure 4, which displays the robust PCA score plot. The ability to analyze whole muscle samples using HR-MAS, without the need for fractionation into polar and apolar components—as required in other experimental approaches—enhances the capacity to discern compositional differences between wild and farmed seabream (Figure 5). (Lines 260-271).
